# Dynamics Differences between Minimal Models of Second and First-Order Chemical Self-Replication

**Lauren A. Moseley** [†,‡,§] and **Enrique Peacock-López** [*,†]

Department of Chemistry, Williams College, Williamstown, MA 01267, USA; laurenm@ldeo.columbia.edu

* Correspondence: epeacock@williams.edu
† The authors contributed equally to this work.
‡ Current address: Lamont Doherty Earth Observatory, Columbia University, Palisades, NY 10964, USA.
§ Current address: Department of Earth and Environmental Sciences, Columbia University, New York, NY 10027, USA.

**Abstract:** To further explore the origins of Life, we consider three self-replicating chemical models. In general, models of the origin of Life include molecular components that can self-replicate and achieve exponential growth. Therefore, chemical self-replication is an essential chemical property of any model. The simplest self-replication mechanisms use the molecular product as a template for its synthesis. This mechanism is the so-called First-Order self-replication. Its regulatory limitations make it challenging to develop chemical networks, which are essential in the models of the origins of Life. In Second-Order self-replication, the molecular product forms a catalytic dimer capable of synthesis of the principal molecular product. In contrast with a simple template, the dimers show more flexibility in forming complex chemical networks since the chemical activity of the dimers can be activated or inhibited by the molecular components of the network. Here, we consider three minimal models: the First-Order Model (FOM), the Second-Order Model (SOM), and an Extended Second-Order Model (ESOM). We construct and analyze the mechanistic dimensionless ordinary differential equations (ODEs) associated with the models. The numerical integration of the set of ODEs gives us a visualization of these systems' oscillatory behavior and compares their capacities for sustained autocatalytic behavior. The FOM model displays more complex oscillatory behavior than the ESOM model.

**Keywords:** chemical self-replication; mathematical model; chemical oscillations; Poincare sections

**MSC:** 34C25; 37G15; 92B05; 92E99

## 1. Introduction

The biochemical networks within living systems are characterized by the synergism of the nonlinear dynamic interactions of their cellular components, including nucleic acids, proteins, membranes, and small molecules. These networks allow for the autonomous growth, adaptation, and passage of information within a molecular ecosystem in response to a continually changing environment. In an attempt to study primordial molecular evolution and the origins of Life, synthetic self-replicating chemical systems have been the focus of numerous theoretical and experimental studies over the past fifty years. The study of self-replicating chemical systems has intensified in the past three decades, shifting from theory to experiment. Past studies of self-replicating systems have utilized nucleic acids [1–8], small organic molecules, and template molecules [9–17], as peptides [18–26], ribozymes, and ribozymes [27–29].

At its most simplified, chemical self-replication is the process by which an individual molecule directs the assembly of component molecules to achieve its duplication. The system is further defined as autocatalytic if the reaction product serves as a specific catalyst

to correctly recognize and position the reactants for future ligation reactions. In an auto-catalytic self-replication system, the product acts as a template to promote the coupling of reactants, and ideally, exponential growth of the product occurs. Several crucial aspects are necessary to optimize the efficiency of a self-replicating system. First, the template must accelerate the reaction rate between the substrates compared to their reaction rate in the absence of the template. Second, the intermediary substrate–template complex must readily form. Third, the resulting template–product complex must readily dissociate to allow for the release and recirculation of the product. The difficulty of the final dissociation is usually responsible for the inability of many self-replicating systems to sustain exponential growth since the number of template molecules available to facilitate self-replication is reduced [30]

One early experimental investigation into the field of chemical self-replication was launched in 1986 by von Kiedrowski [4]. In designing the first self-replicating system of oligonucleotides, von Kiedrowski established that short oligonucleotide palindromes could undergo semi-conservative replication by functioning as templates for ligating short oligonucleotide substrates. In a kinetic analysis of the experimental data, von Kiedrowski et al. [5] found the parabolic growth rate of the autocatalytic system to be proportional to the square root of the concentration of the template or the so-called square root law. In this case, the rate accounts for the influences of autocatalysis and product inhibition in chemical self-replication. Many other biological and synthetic self-replicating systems also yield parabolic growth rates, thus following the square root law and diminishing the Darwinian evolution applicability of these systems.

Julis Rebek, Jr., a pioneer in molecular recognition and self-assembling systems, published several seminal studies on chemical self-replication [9–16]. This work enabled the modeling and mathematical analyses of relevant autocatalytic systems characterized as first-order self-replication [30]. In a 1990 study, Rebek reported a synthetic, self-complementary system in which organic molecules conducted their replication [10]. The autocatalytic reaction saw amino adenosine react with a pentafluorophenyl ester to create a product molecule that combined into a homodimeric template to facilitate subsequent reactions. This work generated excitement for its evolutionary implications and suggested a novel approach to initial thought surrounding the origins of Life, which had assumed fundamental molecular interactions akin to DNA [6–8].

Ghadiri and coworkers published another influential study in 1996 [18], reporting the first experimentally demonstrated instance of autocatalytic peptide self-replication. The self-replicating molecule was a 32-residue $\alpha$–helical peptide based on the leucine-zipper homodimerization domain of yeast transcription factor GCN4, differing by six mutations from the sequence of GCN4. In 15- and 17-residue fragments, the peptide was shown to catalyze its production by accelerating the thioester-promoted amide bond formation of the shorter fragments in a neutral aqueous solution. Chmielewski and colleagues developed self-replicating chemical systems using $\alpha$-helical peptides [23–26]. These peptides could only undergo autocatalytic template formation under acidic conditions due to the necessary protonation of glutamic acid side chains. Like von Kiedrowski's oligonucleotide system, the $\alpha$-helical peptide self-replication of both Chmielewski and Ghadiri groups displayed a parabolic growth rate consistent with the square root of the initial template concentration [19–25].

A 1994 study published by Nicolaou and Li [8] considered the nonenzymatic replication of a 24-monomer palindromic DNA fragment. In the autocatalytic system, a symmetrical polypurine/polypyrimidine duplex acted as a template to ligate two adjacently annealed oligodeoxyribonucleotides in the presence of N-cyanoimidazole. Most uniquely, the reported autocatalytic activity involved forming an intermediate triplex template structure, a stark contrast to the monomeric templating system that characterized prior studies on chemical self-replication.

In 2002, the results of a study conducted on an autocatalytic ribozyme were published by Joyce et al. [27]. A self-replicating system was constructed using the RNA-dependent R3C RNA ligase ribozyme, selected for its simple secondary structure and rapid ligation

rate. Joyce restructured the R3C ligase to design a self-replicating molecule into a symmetrical dimer that would form a product identical to the original template molecule upon undergoing an RNA-catalyzed ligation reaction. Notably, Joyce's modified protein overcame the square root law that had governed all previous studies on autocatalytic chemical networks and exhibited sustained exponential growth [28,29].

By 2004, another viable multimeric template molecule was featured within a synthetic peptide network in the work of Ashkenasy et al. [31], and Ashkenasy's laboratory at Ben-Gurion University of the Negev [32–34] published findings of particular significance. Conclusions indicate that de novo-designed peptide networks can mimic the essential logic functions of complex biological networks. Specific peptides within the synthetic network displayed autocatalytic activity and impressive efficiency as cross-catalytic templates. Most remarkably, the peptide network featured a viable multimeric template molecule essential to developing the second-order self-replication model. These results demonstrated the value of modeling synthetic chemical systems in advancing and better understanding our existing knowledge of complex cellular networks. A 2015 study [35–37] by the same laboratory focused entirely on internetwork competition and cooperation, greatly informing our project on the interactions between competing self-replicating systems.

In the following section, we introduce three generalized self-replicating models to represent a variety of theoretical chemical systems. Beginning with a straightforward Templator model [30], we modify the autocatalytic reaction mechanisms to create more realistic and dynamically richer dimer and trimer models. We consider the nonlinear chemical kinetics of each model in Section 3 using numerical integration, which allows for finding parameters and conditions via the construction of Poincare–Andronov–Hopf (PAH) bifurcation diagrams [38–41]. In Section 4, we compare the oscillator behavior of the Templator and the dimer. We follow the previous analysis with a comparison of the dimer and the trimer in Section 5, including a subsection on the role of template dissociation within the trimer system. The study concludes in Section 6 with a discussion of the results and their implications.

## 2. Minimal Models

In general, proposed reaction mechanisms were developed from previous laboratory work on chemical self-replication to analyze the dynamic behavior of self-replicating systems. The mechanisms were then rendered into mathematical models, in which a set of dimensionless ordinary differential equations (ODEs) is used to describe the change in concentration of a chemical species with respect to time.

### 2.1. Templator Model of First-Order Self-Replication

The first model, the Templator, is a model of a primitive self-replicating molecule based on the research of Rebek et al. [9–16]. The following mechanism schematically represents the model:

$$N + E \xrightarrow{k_u} S \tag{1}$$

$$N + E + S \xrightarrow{k_t} S + S \tag{2}$$

Substrates $N$ and $E$ collide with low probability to form template $S$ in an uncatalyzed initial reaction, with a reaction rate of $k_u$. Once formed, $S$ preferentially binds $N$ and $E$ to form complex ternary NES in an intermediary reaction. The reactive ends of either substrate are placed in close proximity within NES to facilitate the production of another $S$ molecule, a template-mediated reaction with a reaction rate of $k_t$. The SS template–product complex then dissociates into two copies of $S$, and the replication cycle continues. Furthermore, considering stoichiometrically balanced inputs and initial conditions, we can reduce our analysis from a three-variable to a two-variable system. For details see references [42,43].

The following ODEs have been used to analyze the nonlinear dynamics of the Templator system within the chemical pool approximation (CPA):

$$\frac{dX}{dt} = r_o - k_u X^2 - k_t X^2 Y \tag{3}$$

$$\frac{dY}{dt} = k_u X^2 + k_t X^2 Y - R(Y) \tag{4}$$

where $X \equiv (N + E)/2$, and the constant input, $r_o$ characterize an open system in the CPA, and the sink or removal term, $R(Y)$, can model either degradation of the product, or in our case, an enzymatic reaction characterized the Michaelis–Menten (Briggs–Haldane) (MMBH) rate:

$$R(Y) = \frac{V\,Y}{K+Y} \tag{5}$$

which is a Hill function with $n = 1$.

Notice that the second mechanistic step is a pseudo-third-order process, which simplifies a set of bimolecular processes, where the intermediates are fast variables compared with the reactants and products. Therefore, one can use the steady-state approximation (SSA) to eliminate the intermediate fast variable, yielding a pseudo-third-order process.

### 2.2. Minimal Dimer Model of Second-Order Self-Replication

To broaden the scope of the study, the Templator was slightly adjusted to account for self-replicating chemical systems that are directed by an autocatalytic template composed of multiple molecules. Based on the research of Ashkenasy's group, a dimer model is schematically identical to the Templator, except for its dimer template. The following mechanism schematically represents this minimal model:

$$N + E \xrightarrow{k_u} S \tag{6}$$

$$N + E + S + S \xrightarrow{k_t} S + S + S \tag{7}$$

Substrates $N$ and $E$ randomly collide in the initial, uncatalyzed step to form template $S$. Diverging from the previous model; two template molecules associate to form an active autocatalytic dimer (SS) that preferentially binds $N$ and $E$ into a NESS complex in an intermediary step. The substrates are positioned close to each other within the NESS multimer to increase the favorability of the ligation reaction. This multimer dissociates into three $S$ monomers, and the cycle begins again. In the final step, the substrates undergo a template-mediated reaction to produce three template molecules in an SSS dimer–product complex. The following ODEs are used to analyze the minimal dimer system mathematically:

$$\frac{dX}{dt} = r_o - k_u X^2 - k_t X^2 Y^2 \tag{8}$$

$$\frac{dY}{dt} = k_u X^2 + k_t X^2 Y^2 - R(Y) \tag{9}$$

where we have used the same simplifications as in the previous case.

### 2.3. The Extended Model of Second-Order Self-Replication

The final minimal model, the trimer, also accounts for chemical self-replication directed by an autocatalytic template composed of multiple molecules. Unlike the previous two models, however, an additional step is included in the model's mechanism to account

for the association and dissociation of the template dimer. The following mechanism schematically represents the Trimer model:

$$N + E \quad \xrightarrow{k_u} \quad S \tag{10}$$

$$S + S \quad \xrightarrow{k_a} \quad D \tag{11}$$

$$D \quad \xrightarrow{k_d} \quad S + S \tag{12}$$

$$N + E + D \quad \xrightarrow{k_t} \quad S + D \tag{13}$$

Substrates $N$ and $E$ randomly collide in the initial step to form a monomeric product, $S$. In an additional step, two $S$ molecules associate to create a dimeric catalyst, $D$, which acts as a template to bring the substrates together in a multimeric NED complex. Due to the favorable positioning of the substrates within the complex, a ligation reaction occurs to produce an SD complex. To conclude, the intermediary SD product dissociates into three $S$ molecules, and self-replication continues. The following ODEs mathematically represent the trimer:

$$\frac{dX}{dt} \quad = \quad r_o - k_u X^2 - k_t X^2 Z \tag{14}$$

$$\frac{dY}{dt} \quad = \quad k_u X^2 + k_t X^2 Z - R(Y) - 2k_a Y^2 + 2k_d Z \tag{15}$$

$$\frac{dZ}{dt} \quad = \quad k_a Y^2 - k_d Z \tag{16}$$

where we have considered the same simplification as in the previous cases.

In this section, we propose three minimal models of self-replication, which we analyze in the following section. The first two models consist of two variables, which implies that one could obtain analytical results for the steady states (SS) and the stability of these SS. Most of these analytical results can be found elsewhere [44]. In contrast, the third model includes one more mechanistic step, the formation of a stable catalytic dimer. Consequently, we end with a three-variable system which implies that we have to deal with a cubic equation that is hard to analytically dissect, but numerical analyses are easy to perform.

## 3. Dynamic Characterization

In this section, we want to compare the two-parameter bifurcation diagram for the three minimal models and understand the effect of the different nonlinearities in parameter space. For the two-variable case, it is well known that we only have to consider the Jacobian trace to determine the SS's stability. For the three-variable case, we consider the real part of the eigenvalues associated with the Jacobian. In this case, the zeros of the $Re(\lambda)$ give us information about the Poincare–Andronov–Hopf (PAH) bifurcation diagrams [38–41].

The concentrations of substrates $N$ and $E$ are mathematically converted into the variable $X$ and are assumed to be of equal initial concentrations for the analytical purposes of this study. The concentration of the template molecule is represented by variable $Y$, which denotes the template monomer of the FOM and the template dimer of the SOM. Of particular note are parameters $K$ and $r_o$. The Michaelis constant, $K$, is specific to the catalytic template molecule of each self-replicating system and is used to characterize the enzymatic sink of each autocatalytic reaction. The $r_o$ parameter represents the input of reactants N and E into the chemical pool of each self-replicating system, which can be externally controlled.

While the FOM and the SOM share the same parameters, the ESOM contains several additional parameters. Unlike the FOM and SOM, the variable Y represents the concentration of the monomeric template product $S$ as shown in Equation (10) of the ESOM mechanism. Consequently, the variable Z is added to represent the concentration of the dimeric template D formed by the association of two $S$ products as depicted in Equation (11). Two parameters, $k_a$ and $k_d$, distinguish the duplex formation, as the pres-

ence of these values incorporates the association and dissociation of the dimeric template molecule into a computational analysis of the system's mathematical mechanism.

Regardless of these additional parameters, the ESOM shares a cubic nonlinearity with the FOM. The association of three molecules determines the nonlinearity to form the template product, as shown in each model's drafted mechanisms. The SOM has a quartic nonlinearity, as the model's template product is formed by associating four molecules. Conclusions from an analysis of the SOM are compared to those of the FOM and ESOM in two separate sections. This analysis was conducted at varying values of $k_u$ to represent a slower ($k_u = 0.01$) and slightly faster ($k_u = 0.10$) rate of the initial, uncatalyzed reaction. Without an autocatalytic step, chemical oscillations are believed to not be observed.

In our analyses, we consider that the ODEs, associated with each model are dimensionless, where $k_t$ and $V$ are equal to unity. For a detailed discussion of our scaling, see the reference [42]. In this case, $k_u < 1$ and $r_o < 1$, and we consider $r_o$, $K_M$, $k_a$, and $k_d$ as our bifurcation parameters. Based on this assumption, each model's efficacy as a self-replicating system will be determined through an examination of the oscillatory behavior reflected in PAH bifurcation diagrams. These diagrams are constructed using the following exact transcendental equation, which we derived for a generalized minimal two-variable model of self-replication:

$$K = \frac{(1-r_0)\left[(n-1+r_o) - k_u \frac{(1-r_o)^{n+1}}{K^n\, r_o^n}\right]^{\frac{m}{m+n}}}{m^{\frac{m}{m+n}}\, r_o^{\frac{m+n-1}{m+n}}\left[1 + k_u \frac{(1-r_o)^n}{K^n r_o^n}\right]^{\frac{n+1}{m+n}}} \tag{17}$$

where $m = 2$ for both FOM and SOM, but $n = 1$ for FOM, and $n = 2$ for SOM. The relation in Equation (17) yield the values of $K$ and $r_o$ that yield steady states with a vanishing real part of the eigenvalue related to the Jacobian calculated at the steady state, $Re\lambda(K, r_o) = 0$. For details, see reference [30].

For two-variable models, the trace of the Jacobian calculated at the steady states is equivalent to the real part of the eigenvalues, $TrJ = Re\lambda$. Therefore in Figure 1, we depict, in parameter space, the behavior of the real part of the eigenvalue associated with the Jacobian calculated at steady states. The steady states depend on the parameter values, $K$ and $r_o$ since we fixed $k_u$. The thick closed curve, $K(r_o)$, represents the values where the $Re\lambda = 0$, and where the $Re\lambda$ changes sign from negative, outside the curve, to positive, inside the curve, while the imaginary part of the eigenvalues is nonzero. Therefore, the curve $K(r_o)$ is a sort of a PAH in two-dimensional parameter space.

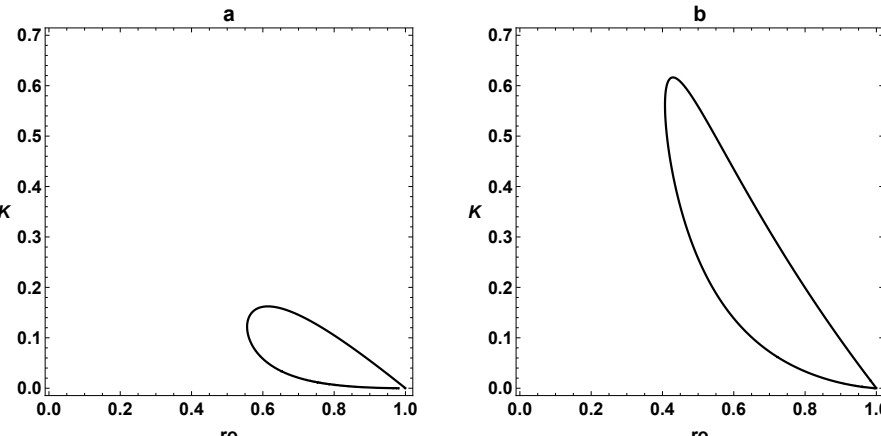

**Figure 1.** Poincare–Andronov–Hopf (PAH) bifurcation diagrams of $K$ vs. $r_o$ for the (**a**) First-Order- and (**b**) Second-Order Models under the parameter of $k_u = 0.10$.

This observation indicates that the system is capable of stable oscillations, characteristic of autocatalysis. As the loop increases in size, the probability of autocatalytic behavior in

the system increases because more parameters will produce sustained chemical oscillations. Two sets of $(r_o, K)$ parameters were derived from these bifurcation diagrams to compare viable oscillatory solutions between the two systems. In the first set, the chosen parameters fall within the bifurcation loops of both models. In the second, the selected parameters fall outside of the FOM or ESOM loops and inside of that of the SOM. The range of $r_o$ values that fall within the bifurcation loop is particularly significant to wet-lab experimentation since $r_o$ can be externally controlled and the enzymatic $K$ constant cannot. This $r_o$ range will be the gauge by which the models are compared and contrasted.

### 3.1. First- and Second-Order Minimal Models of Self-Replication

The first comparison between the FOM and SOM focuses on the differences between the model's oscillatory behavior as $k_u$ is set to 0.1. The resulting PAH bifurcation diagrams under this condition are displayed in Figure 1. When scaled to the same graphical dimensions, $r_o = (0.0, 1.0)$, $K = (0.0, 0.7)$, the loop of the SOM's bifurcation diagram is considerably larger in area than that of the First-Order Model. The SOM's loop spans a range of $K$ values nearly four times greater than the FOM, reaching a value of 0.6 while the FOM extends to 0.15.

As evidenced in Figure 2, the SOM system displays more robust oscillations. Under both sets of conditions, the FOM shows a comparatively minimal amount of oscillatory behavior, ultimately leading to the system's steady state. In Figure 2a, a PAH bifurcation diagram constructed using a parameter set that falls within the model's bifurcation loop displays only several transient oscillations before dying out. Similarly, in Figure 2c, the system exhibits a single oscillatory perturbation followed by a return to a steady state. In direct contrast, the SOM's oscillations are robust under both conditions. The system's oscillations are most distinctive in Figure 2b, with oscillations of identical amplitudes of 1.6 and wavelengths of 10. In Figure 2d, the transient oscillation is larger in amplitude than subsequent oscillations, but the system's activity remains robust. Also, in Figure 3, we plot the trajectory for different initial conditions showing either the spiral in or the spiraling out toward the attractor.

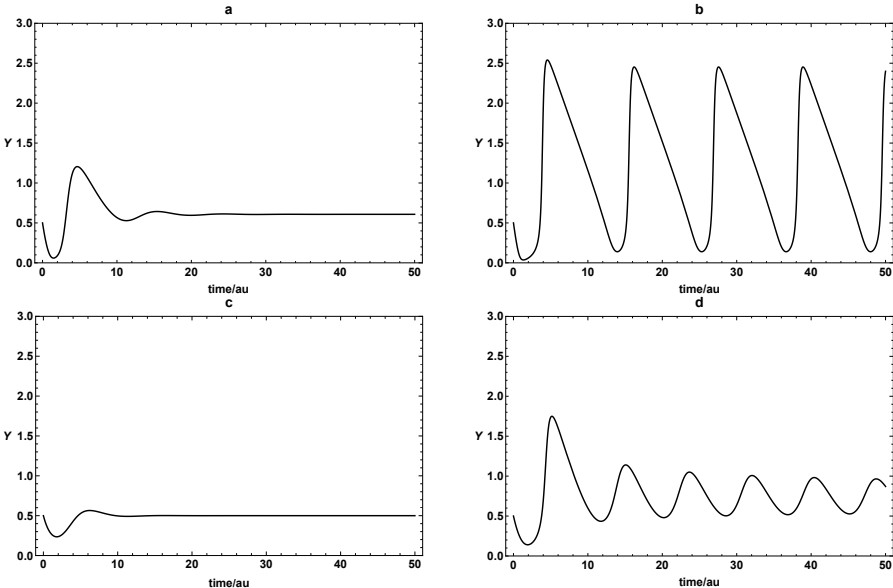

**Figure 2.** Examples of different dynamic behaviors of the concentration of the product $Y(\tau)$ for the same parameter values for the First-Order and Second-Order Models. We chose parameters using the PAH bifurcation diagrams for $k_u = 0.10$. (**a**) FOM, $r_o = 0.70$, and $K = 0.15$. (**b**) SOM, $r_o = 0.70$, and $K = 0.15$. (**c**) FOM, $r_o = 0.50$, and $K = 0.50$. (**d**) SOM, $r_o = 0.50$, and $K = 0.50$.

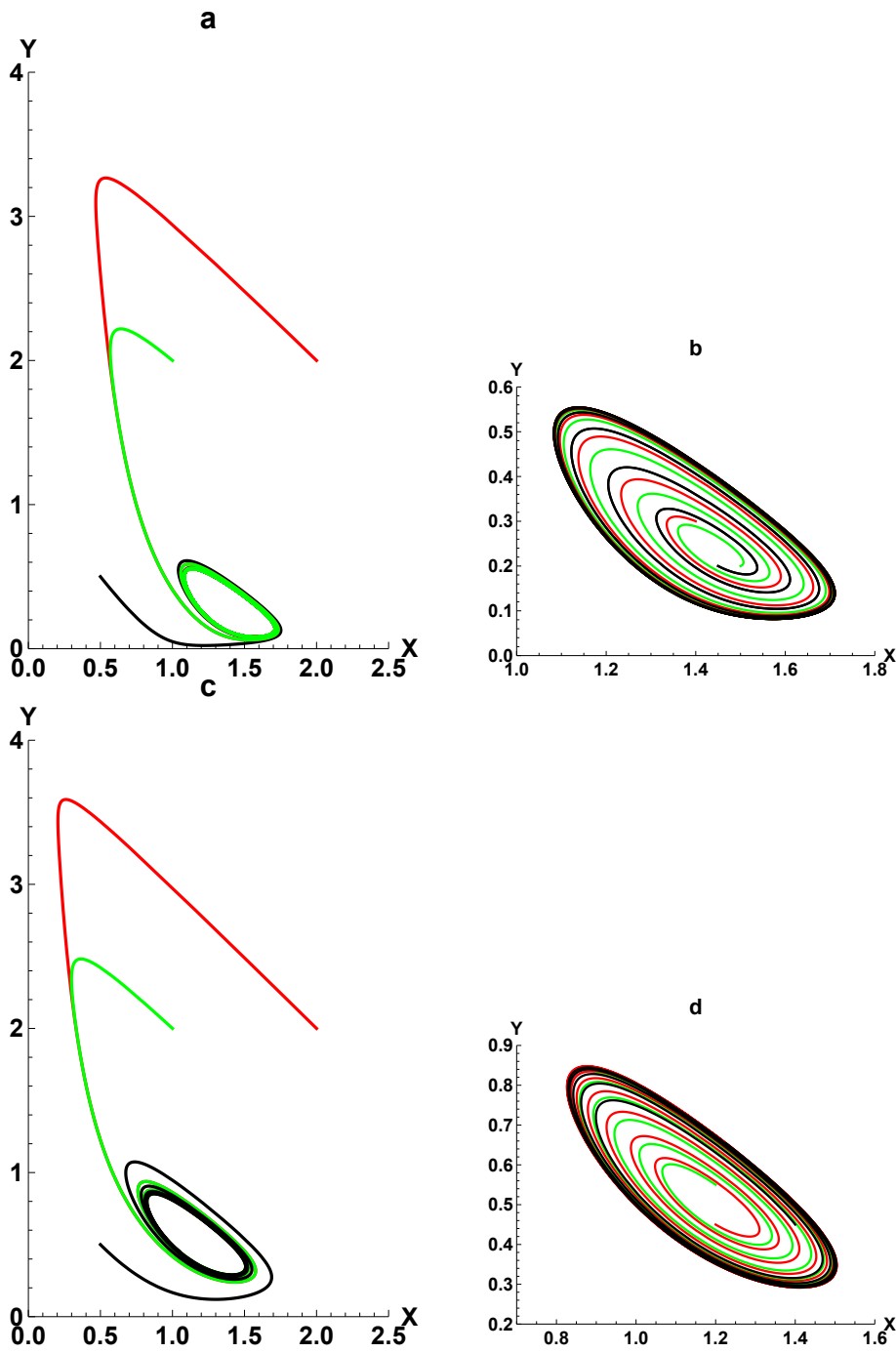

**Figure 3.** Examples of different chemical oscillations of the concentration in phase space $(X, Y)$. We chose parameters using the PAH bifurcation diagrams as follows: (**a**,**b**) FOM, $k_u = 0.10$ $r_o = 0.70$, and $K = 0.10$; (**c**,**d**) SOM, $k_u = 0.10$ $r_o = 0.50$, and $K = 0.50$. Colors represent different initial conditions (IC) for (**a**,**c**) external IC, (**b**,**d**) internal IC.

In another comparison between the FOM and SOM models, $k_u$ is set to 0.01, and the resulting bifurcation diagrams shown in Figure 4 are constructed using Equation (7). The change in the value of the uncatalyzed reaction rate makes a dramatic impact on the range of autocatalysis-inducing parameter sets for the SOM. Extending from $K = 0.6$ to $K = 9.5$, the SOM displays an approximately 16-fold increase in parameter range under $k_u = 0.1$. The First-Order system is minimally affected by the change in $k_u$, increasing from 0.15 to 0.9 in the range of $K$ values contained within the bifurcation loop. When

appropriately scaled to equal dimensions, it is apparent that the FOM's self-replicating parameters are limited compared to those of the SOM.

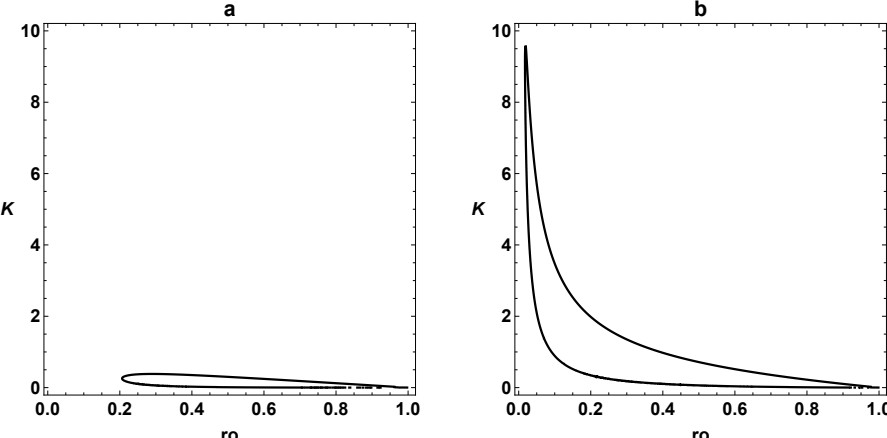

**Figure 4.** PAH bifurcation diagrams of $K$ vs. $r_o$ for the (**a**) First-Order and (**b**) Second-Order Models under the parameter of $k_u = 0.01$.

Using parameters selected from the previous bifurcation diagrams, we visualize the dynamic differences between the FOM and the SOM in the examples of chemical oscillations shown in Figure 5. Following the slightly expanded parameter range of the FOM, the model displays sustained oscillations in Figure 5a under parameters that fall within its bifurcation loop. As shown in Figure 5b, oscillations with a larger amplitude and wavelength appeared when the same parameters were used for the SOM. Figure 5c,d illustrate the enormous oscillatory range of the SOM compared to the limited range of the FOM. Only the SOM exhibits oscillations under parameters extracted from the Second-Order's bifurcation loop. Almost identical to Figure 2c, the FOM experiences a single oscillatory perturbation followed by a relatively quick return to a steady state.

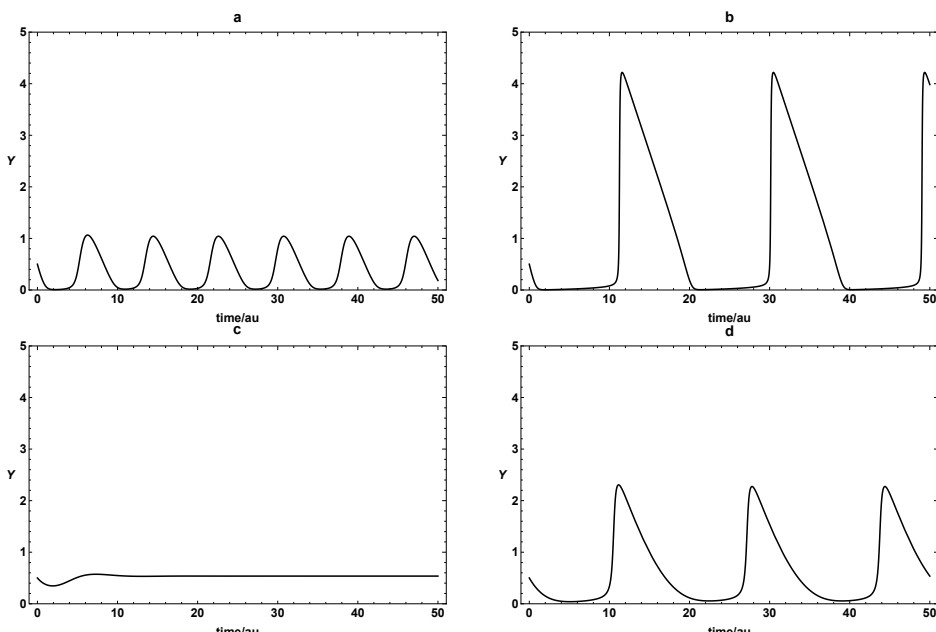

**Figure 5.** Examples of different dynamic behaviors of the concentration of $Y(\tau)$ for FOM and SOM under parameters defined by the PAH bifurcation diagrams, $k_u = 0.01$. (**a**) First Order, $r_o = 0.5$, and $K = 0.20$. (**b**) Second Order, $r_o = 0.50$, and $K = 0.20$. (**c**) First Order, $r_o = 0.30$, and $K = 1.0$. (**d**) Second Order, $r_o = 0.30$, and $K = 10$.

### 3.2. Extended Second-Order Self-Replication

Unlike the SOM, the ESOM incorporates a mechanistic step to represent the association and dissociation of the dimeric template. As a result, linear stability analysis of the model must include two more parameters: $k_a$ and $k_d$. As displayed in Figure 1, the most suitable values of $k_a$ and $k_d$ were determined by holding all other parameters constant ($k_u = 0.01, r_o = 0.5, K = 0.1$) and selecting which values of $k_a$ and $k_d$ best optimize ESOM oscillatory behavior. To thoroughly examine the relationship between $k_a$, $k_d$, and oscillatory behavior, a variety of conditions were constructed in which $k_a > k_d$, $k_a < k_d$, and $k_a = k_d$. The results of these parameter manipulations are shown in Figure 6.

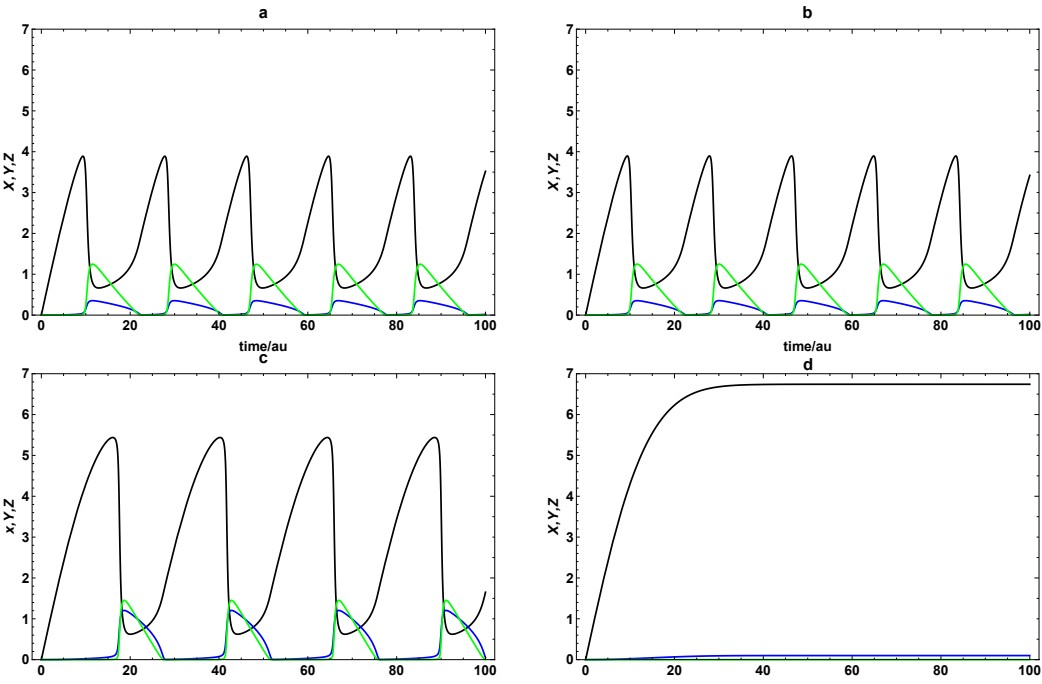

**Figure 6.** Examples of ESOM dynamic behaviors under constant parameter values ($k_u = 0.01$, $r_o = 0.5$, $K = 0.1$) and varying values of $k_a$ and $k_d$. (**a**) $k_a = 10,000$; $k_d = 1000$. (**b**) $k_a = 1000$; $k_d = 100$. (**c**) $k_a = 1000$; $k_d = 1000$. (**d**) $k_a = 100$; $k_d = 1000$. (X = black, Y = blue, Z = green).

The identical amplitudes and wavelengths of the oscillations of Figures 5b and 6a indicate that the most critical aspect of $k_a$ and $k_d$ to the resulting chemical oscillations is their ratio. Both sets of parameters contain a 10:1 ratio of $k_a$ to $k_d$. When the ratio is 1:1, as in Figure 6c, the resulting oscillatory behavior is largely the same in its oscillatory properties. The only oscillatory change concerns the first full oscillation—this amplitude is smaller than the initial peaks found in the first two examples. Still, this transient behavior is of no interest. Oscillations do not occur when $k_d$ is higher than $k_a$, as shown in Figure 6d. Due to the strength of the oscillations under the conditions found in Figure 6a,b, a 10:1 ratio was incorporated into the following analysis of the Extended Second-Order. For completion in Figure 7, we depict trajectories yielding oscillations in $(X, Y, Z)$ phase space projected on the $(X, Y)$ plane for a particular set of parameter values and different IC.

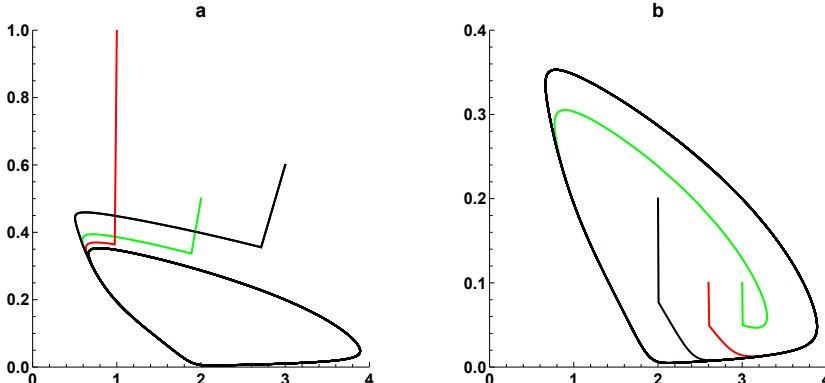

**Figure 7.** Examples of ESOM oscillations in phase space $(X, Y)$, using for the following parameter values: $k_u = 0.01$, $r_o = 0.50$, $K = 0.10$, and $k_a = 10{,}000$, and $k_d = 1000$. Colors represent different initial conditions (IC) for (**a**) external IC and (**b**) internal IC. Notice that path crossing is due to the projection of 3D $(X, Y, Z)$ onto 2D $(X, Y)$.

*3.3. Comparison between SOM and ESOM*

In the first comparison between the autocatalytic efficacies of the SOM and ESOM models, $k_u$ is set to 0.10, and the resulting PAH bifurcation diagrams are displayed in Figure 8. When scaled to the same dimensions, the loop of the SOM's bifurcation diagram has a significantly greater area than that of the ESOM. The ESOM bifurcation diagram resembles a smaller, flatter version of the SOM's loop, with a minimal range of oscillation-producing $K$ values reaching barely 0.10. Under the same conditions, the SOM exhibits viable parameter sets featuring $K$ values nearly seven times the height of the ESOM's loop and spans up to 0.60.

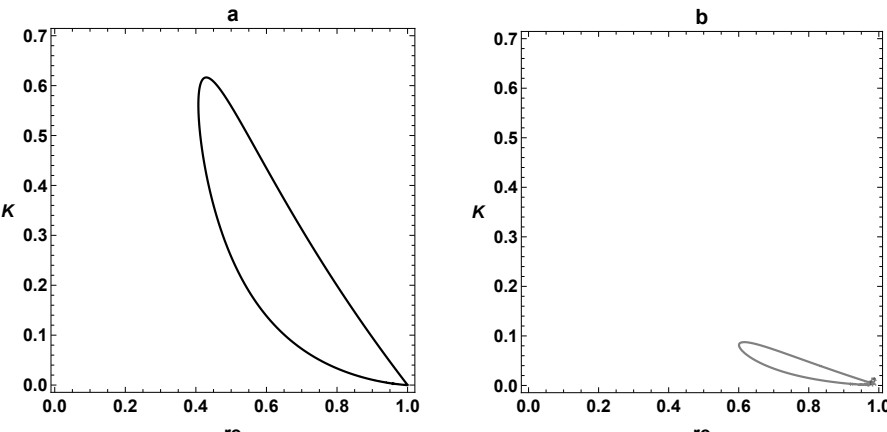

**Figure 8.** Two parameters, $K$ vs. $r_0$, PAH bifurcation diagrams with $k_u = 0.10$ for (**a**) SOM and (**b**) ESOM with $k_a = 500$, and $k_d = 100$.

As shown in Figure 9, the SOM system displays more robust chemical oscillations under both sets of parameters derived from the previous bifurcation diagrams. The difference in oscillatory behavior in Figure 9a,b is evidence for this observation. Under a set of parameters selected to create oscillations in both systems, the SOM produces a larger amplitude of 2.5 and a shorter wavelength of 35. The first oscillation produced by the ESOM is sluggish in comparison, with an amplitude of 1.9 and a wavelength of 93. Moreover, the significantly decreased amplitude—0.8—of the subsequent ESOM oscillation indicates that the system is approaching its steady state. This observation directly contrasts the SOM, which Figure 9c implies has a continuous and consistent oscillatory system over the ESOM's return to a steady state in Figure 9d.

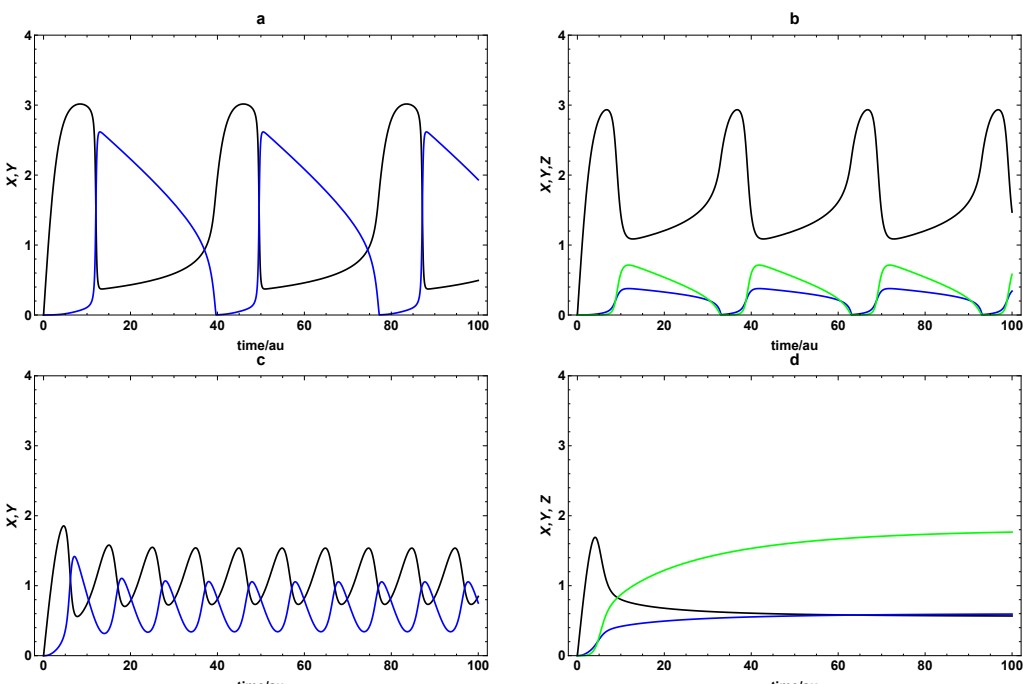

**Figure 9.** Examples of dynamic behaviors for the SOM and ESOM under parameters determined using the PAH bifurcation diagrams, $k_u = 0.1$, $k_a = 500$, $k_d = 100$. (**a**) SOM, $r_o = 0.95$, and $K = 0.005$. (**b**) ESOM, $r_o = 0.95$, and $K = 0.005$. (**c**) SOM, $r_o = 0.60$, and $K = 0.4$. (**d**) ESOM, $r_o = 0.60$, and $K = 0.4$. ($X$ = black, $Y$ = blue, $Z$ = green).

In the final comparison of the two models, the value of $k_u$ is lowered to 0.01 to simulate a slower uncatalyzed reaction within either system. The alteration produces dramatic results for both the systems, substantially increasing the $K$ range for each model's PAH bifurcation diagrams as displayed in Figure 10. The SOM's $K$ range grows from 0.6 to 9.5, increasing by a factor of 16 from the bifurcation diagram constructed under $k_u = 0.1$ in Figure 8. The ESOM's loop increases its $K$ range from 0.1 to 3—a 30-fold increase—to resemble a flatter, smaller version of the SOM. Although the scale of the increase in range is more significant for the ESOM, the SOM's loop has the largest area. It thus contains more parameter sets that describe viable conditions for oscillatory behavior.

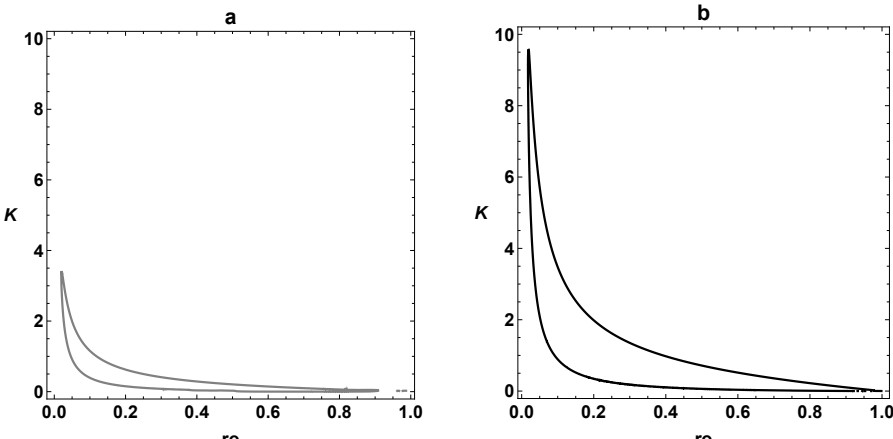

**Figure 10.** Two parameters, $K$ vs. $r_0$, PAH bifurcation diagrams with $k_u = 0.010$ for (**a**) SOM and (**b**) ESOM with $k_a = 500$, and $k_d = 100$.

The autocatalytic behavior of the two models is differentiated by the following set of parameters drawn from the interior of the SOM's bifurcation loop. As shown in Figure 11, the SOM demonstrates stronger chemical oscillations under both sets of parameters. When

parameters extracted from the PAH bifurcation diagrams to generate oscillations in both systems are applied to the models, the SOM demonstrates an oscillation with the largest recorded amplitude, 6.5, out of all the listed oscillatory trials. The ESOM also performs well, displaying robust oscillations with an amplitude of 2 and a wavelength of 33. In Figure 11c, the SOM exhibits stable oscillations that are characterized by an average amplitude of 1.7 and a wavelength of 20. No oscillations occur under these conditions for the ESOM, which attains its steady state in Figure 11d.

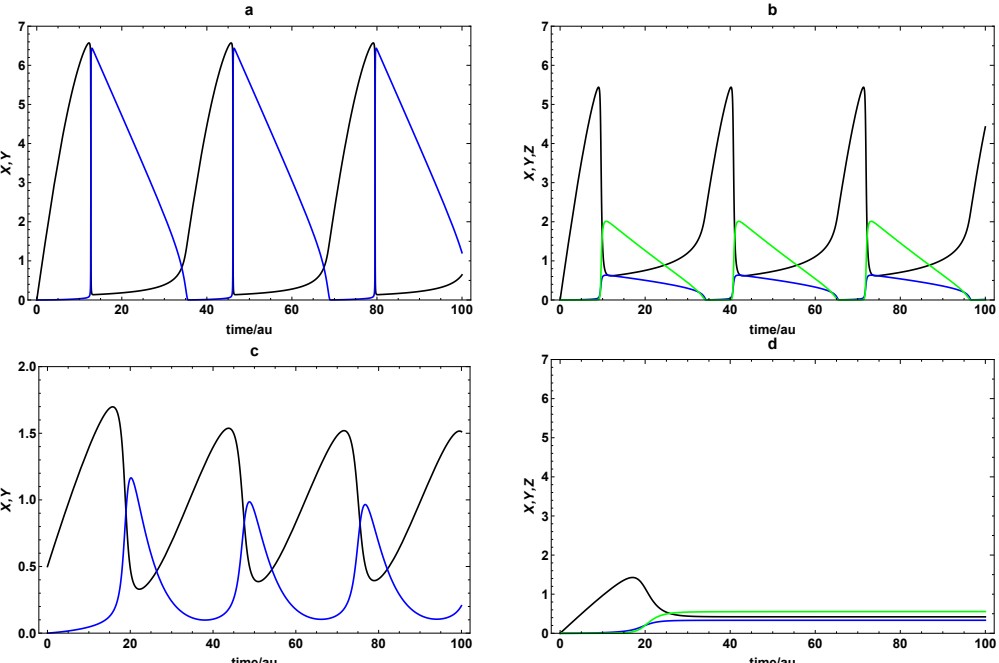

**Figure 11.** Examples of dynamic behaviors for the SOM and ESOM under parameters determined by the PAH bifurcation diagrams, $k_u = 0.010$, $k_a = 500$, $k_d = 100$. (**a**) SOM, $r_o = 0.75$, and $K = 0.05$. (**b**) ESOM, $r_o = 0.75$, and $K = 0.05$. (**c**) SOM, $r_o = 0.10$, and $K = 3$. (**d**) ESOM, $r_o = 0.10$, and $K = 3$. ($X$ = black, $Y$ = blue, $Z$ = green).

## 4. Dynamic Characterization

To further characterize the effect of the nonlinearities, we consider the maximum–minimum Poincare section for the $Y$ component. In Figure 12, we depict the Poincare section for FOM for $K = 0.20$ and recall that in our scaling, $k_t = V = 1.0$. The numerical analysis considers that as we change the parameter value $r_o$, the steady states, ($Re(\lambda) < 0$), go through a bifurcation, ($Re(\lambda) = 0$), changing to unstable steady states, ($Re(\lambda) < 0$), to yield stable limit cycle.

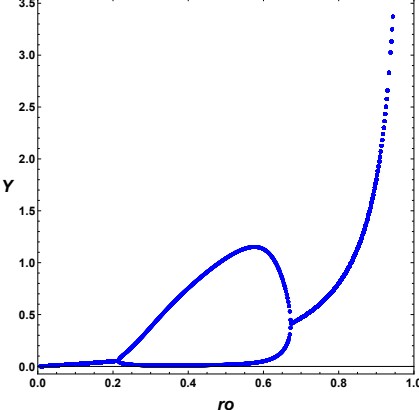

**Figure 12.** Max–min Poincare section for FOM with $k_t = 1$, $V = 1$, $k_u = 0.01$, and $K = 0.20$.

In contrast, Figure 13 depicts the same bifurcation diagram for SOM and ESOM. For these two models, $k_u = 0.01$, $K = 0.20$, and $k_a = 100$, $k_d = 100$.

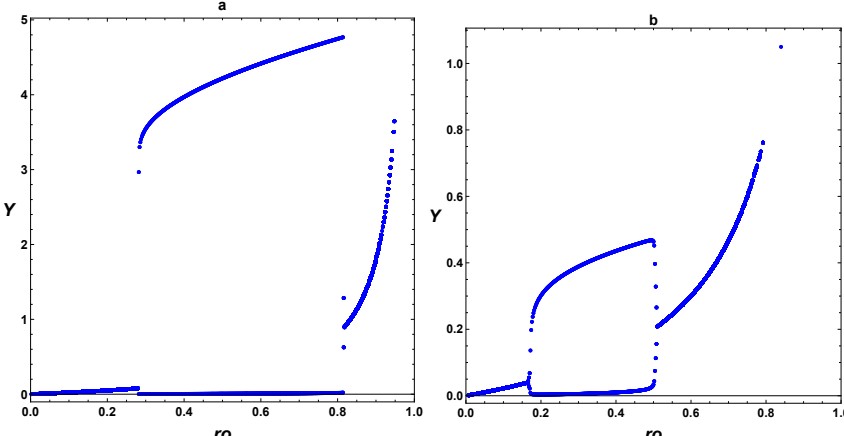

**Figure 13.** Max–min Poincare section for (**a**) SOM and (**b**) ESOM with $k_t = 1$, $K = 0.20$, $k_a = 500$, and $k_d = 100$.

Notice a significant chance between Figure 12a (FOM) and Figure 13a (SOM), where the quartic nonlinearity changes the "soft" to a "hard" bifurcation. The so-called canard is depicted in Figure 14. Also, the amplitude of the oscillations increases considerably. But, in the case of ESOM, the amplitude is reduced. The reduction in amplitude in the case of ESOM is expected because the values of $h_a$ and $k_d$ favor the formation of the dimer (Z) at the expense of the monomer (Y), but the bifurcation is similar to a canard.

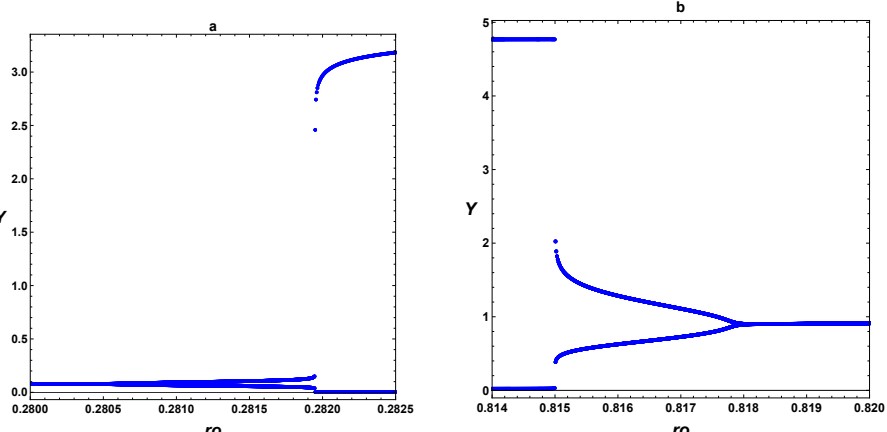

**Figure 14.** Max–min Poincare section for SOM with $k_t = 1$, $K = 0.20$, (**a**) small values of $r_o$, (**b**) larger values of $r_o$.

The presence of a canard due to the quartic nonlinearity may be relevant for minimal models, but it may disappear if the mechanism is extended beyond our ESOM. However, as an example of "hard" transitions, it is worth showing them in more detail in Figure 15, where one can see the transition from small amplitude oscillations, P2, to large amplitude oscillations, P2. The studies of canard transitions have been linked to mixed-mode oscillations and could be relevant for bursting oscillations.

Finally, we characterize the ESOM by the real part of the Jacobian's eigenvalues.

$$
\begin{pmatrix}
\dfrac{2\sqrt{r_o}\sqrt{k_d\,k_u(1-r_o)^2+k^2k_a r_o^2}}{\sqrt{k_d}(1-r_o)} & 0 & -\dfrac{k_d(1-r_o)^2 r_o}{k_dk_u(1-r_o)^2+k^2k_a r_o^2} \\[2ex]
-\dfrac{2\sqrt{r_o}\sqrt{k_dk_u(1-r_o)^2+k^2k_a r_o^2}}{\sqrt{k_d}(1-r_o)} & -\dfrac{1+\left(-3+4k^2k_a\right)r_o+3r_o^2-r_o^3}{k(1-r_o)} & k_d\left(2+\dfrac{(1-r_o)^2 r_o}{k_dk_u(1-r_o)^2+k^2k_a r_o^2}\right) \\[2ex]
0 & \dfrac{2kk_a r_o}{1-r_o} & -k_d
\end{pmatrix}
$$

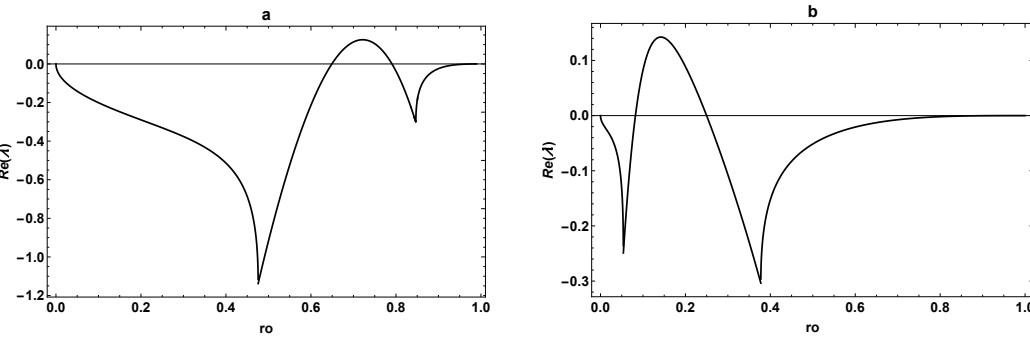

**Figure 15.** $Re(\lambda)$ vs. $r_o$ for ESOM with $K = 0.20$, $k_a = 500$. $k_d = 100$, and (**a**) $k_u = 0.100$, (**b**) $k_u = 0.010$.

In the case of ESOM, we have a three-dimensional system, which implies a cubic equation for the eigenvalues. Although there is a simplification to the cubic equation, numerically, we can determine the eigenvalues and plot them in Figure 15 for two values of the uncatalyzed reaction constant, $k_u$.

To compare the eigenvalue figures, we reproduce the two-parameter bifurcation diagrams in Figure 16, where one clearly can compare the region of the positive real part with the interior of the two-parameter diagram.

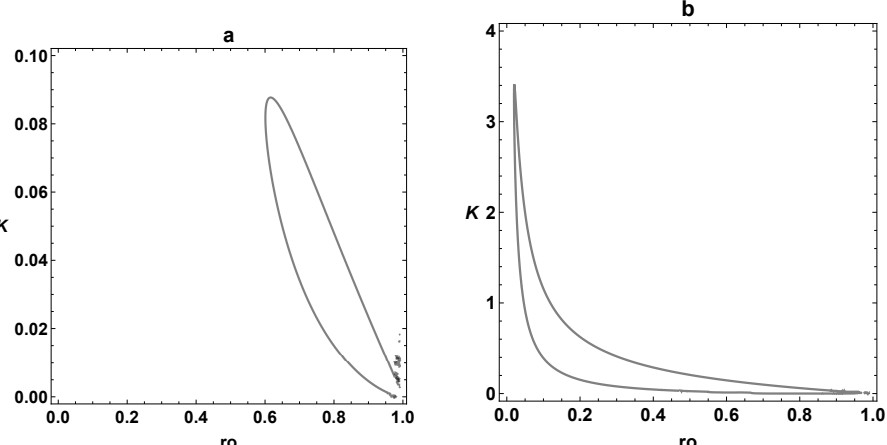

**Figure 16.** Two parameters, $K$ vs. $r_0$, PAH bifurcation diagrams for ESOM with $k_a = 500$, and $k_d = 100$, and (**a**) $k_u = 0.100$, (**b**) $k_u = 0.010$.

Although we have not discussed the steady-state solutions in depth, the solution of the steady-state relations is relatively easy to obtain, and for the FOM and SOME, we have analyzed them elsewhere [30–44]. We have to consider bifurcation diagrams to compare the effect of the nonlinearities and the extension of the model, where we extend the mechanism, including another dynamic variable. Therefore, we can understand the changes due to the usual contraction using the SSA. Indeed, we can contract the ESOM, assuming the SSA for the dimeric species, yielding the minimal SOM.

## 5. Discussion

In Figure 1, we depict the oscillatory region in parameter space ($K, r_o$) and notice that it expands for the SOM compared to the FOM. Notice that the SOM considers a contraction from three variables to two variables, using the steady-state approximation for the dimer, which stabilizes the system, increasing the region in parameter state. In contrast, considering the three-variable model, we have another kinetic parameter describing the dimerization reaction. The parameter, if it favors the formation of the dimer, stabilizes the oscillatory behavior and expands the region in parameter space. But, in contrast, if the parameter favors the template or monomer, the region in the parameter space shrinks. This behavior seems reasonable because the dimer is the catalytic species that is fundamental for the autocatalytic cycle of the mechanism. Finally, if we consider more mechanistic steps, we end up with more kinetic parameters, and depending on their values, we may find an even larger region in parameter space or a smaller one. At least for self-replicating peptides, it will depend on the kinetic behavior of the dimer. Experimentally, in principle, one could test the role of dimerization by considering inhibitory reagents.

From Figures 1 and 4, we can notice the change in parameter space when we change the cubic nonlinearity (FOM) to a quartic nonlinearity (SOM). As it is evident, the increase in area in parameter space implies more parameter value choices but also can be interpreted as the destabilization of the FOM dynamics, as illustrated by Figures 2 and 5. In contrast, the inclusion of the dimer as the catalytic reagent reduces the area in parameter space, clearly depicted in Figures 8 and 10. Therefore, the inclusion of more mechanistic steps stabilizes the system, Figures 8 and 10, but keeping the amplitude and the period the same, as illustrated in Figures 9 and 11. Therefore, one has to be careful when using the SSA approximation to reduce the number of dynamic variables without specific experimental results.

To determine the role of the dimerization dynamics, in Figure 6, we compare the time series for different values of $k_a/k_d$. From the Figure, we can conclude that the concentration of the model via an SS, may be questionable if one wants to compare our results with experimental results. We can still consider dynamic reductions to study possible dynamic behaviors, and one may even be able to obtain analytical results. Still, one may miss a detailed analysis of the parameter space.

In Section 4, we show in Figures 12–14 the max–min Poincare section to illustrate the effect of the nonlinearities on the amplitude of the oscillations. We notice that the quartic nonlinearity has a relevant effect on the amplitude, and it changes a "small" amplitude PAH to a "hard" PAH of the so-called canard type. Notice that if we change the cubic nonlinearity to a quadratic nonlinearity, we still observe an oscillatory dynamic behavior of the so-called Higgins model.

## 6. Conclusions

In general, modeled representation of the origin of Life requires molecular components that show self-replication before one can link the self-replicating network to self-assembly and metabolic network of molecular components. Recently, self-replicator in open systems has attracted the attention of both experimentalist and theorist to consider self-replication in open systems, like the continuous stirred tank reactor (CSTR), because living systems evolve in open systems and settle in nonequilibrium states, which include concentration oscillations [45–48]. For example, Otto's group [49,50] considered dynamic combinatorial libraries (DCLs) of macrocycles in CSTR. Also, in the case of supramolecular self-replicators, the Fletcher group [51] reported autonomous oscillation in the concentration of the self-replicator. Addressing the selection of the fittest, Ashkenasy and de la Escosura [52] studied nucleopeptides.

A comparison of the FOM and SOM reveals the SOM is more likely to exhibit autocatalytic activity, evidenced by the PAH bifurcation diagrams in Figures 1 and 4 and the examples of chemical oscillations displayed in Figures 2 and 5. The FOM's mechanistic backbone alteration to include a dimeric template molecule considerably alters the system's

dynamics. The SOM contains a more extensive set of ($r_o$, $K$) parameter values that can combine to produce chemical oscillations. Furthermore, the variation of $k_u$ from 0.10 to 0.01 solely impacted the oscillatory efficacy of the SOM. The results of this analysis reflect the limited applicability of the FOM. Due to its oversimplified mechanism, the model does not provide a supportive environment for more nuanced autocatalytic activity. An analysis of the role of dissociation within the ESOM's mathematical model was necessary to evaluate the system efficiently. Oscillations appear most prominently when the association rate ($k_a$) outnumbers the dissociation rate ($k_d$). The ratio of 10:1 favors the formation of the catalytic duplex. Similar dynamic results occurred when the ratio was set to 1:1, representing a dimeric template that forms and dissociates in equal measures. Predictably, no oscillations occurred when the $k_d$ was higher than the $k_a$—a trial meant to simulate a template complex that dissociates more quickly than it forms. In a second analysis featuring the ESOM, the SOM again displays the most dynamic potential for self-replication. Unlike the FOM, both the SOM and ESOM's range of viable oscillatory parameter sets was similarly expanded by the lowering of $k_u$ to 0.01. Though not as impacted as the SOM, there was a 30-fold increase in the range of $K$ values included within the ESOM's bifurcation loop. Despite this jump, the SOM continued to dominate and produced bifurcation loops shown in Figures 8 and 10 with larger areas and more robust chemical oscillations, as displayed in Figure 10. Most notably, this investigation indicates that adding detail to an autocatalytic mechanism does not necessarily improve the chances of this model achieving self-replication. While the ESOM was constructed to be more aware of the intricacies of the template's behavior, it is a dynamically weaker system that presents a limited window for chemical oscillations compared to the SOM.

**Author Contributions:** Conceptualization, E.P.-L.; methodology, E.P.-L.; software, L.A.M. and E.P.-L.; validation, L.A.M. and E.P.-L.; formal analysis, L.A.M. and E.P.-L.; investigation, L.A.M. and E.P.-L.; resources, E.P.-L.; writing—original draft preparation, L.A.M. and E.P.-L.; writing—review and editing, L.A.M. and E.P.-L.; supervision, E.P.-L.; project administration, E.P.-L.; funding acquisition, E.P.-L. All authors have read and agreed to the published version of the manuscript.

**Funding:** The National Science Foundation funded this research grant CHE-0911380.

**Data Availability Statement:** The data supporting this study's findings are available from the corresponding author upon reasonable request.

**Acknowledgments:** The authors would like to thank Nathaniel Wagner and Gonen Ashkenasy for the uncountable discussions on chemical self-replication.

**Conflicts of Interest:** The authors declare no conflict of interest.

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
