# Peer review of "Dynamics Differences between Minimal Models of Second and First-Order Chemical Self-Replication"

_2673-8716, doi:10.3390/dynamics3030023_

Round 1
Reviewer 1 Report
The paper compares results from several different types of minimal model describing self-replication processes. There a few minor points for the authors to consider:
The abstract could do with a few more lines of context i.e. some discussion on how these models related to origin of life and what is meant by a first order and second order model to better explain to the reader why this work was carried out- why compare these types of models?
The “colleaguescite” on page 2 should just be colleagues?
There is something wrong with the numbering figure 2 (two bs). Actually this looks like only damped oscillations in the FOM with ku = 0.1, i.e. stable focus – are these parameters correct for oscillations, perhaps better to compare for slightly lower K?
In the discussion, could there be more of an explanation in the context of the kinetics and mechanism that may help to understand as to why the region of oscillations is so much greater with the SOM, to expand on the point that inclusion of more steps stabilises oscillations? This may be useful for those trying to obtain robust oscillations in models.
.
There are a few minor corrections
Author Response
Attached PDF

Reviewer 2 Report
Referee report on "Dynamics differences between minimal models of second and first-order chemical self-replication" by Lauren A Moseley and Enrique Peacock-Lopez The authors discuss three models related to self-replication and reaction kinetics. All models are given by simple autonomous systems of ODEs, which are studied numerically. From the presented numerical results we can see that oscillations are possible in the systems. The study presented in the paper is of some interest, however the paper is careless written. For instance, equation (4) does not make sense, the title of subsection 2.1 is written as "Templator Model of first0order self-replication", periods at the end of some sentences are missed, it is not clear what is plotted in Figure 2 etc. My main concern is the following. The authors draw some diagrams which they call Poincare-Andronov-Hopf bifurcation diagrams. However, to my understanding, in Figure 1 they just found a set in the space of the parameters where the Jacobian at the system’s steady state is positive. This is not a bifurcational diagram in the usual sense (see e.g. [Yu. Kuznetsov, Elements of Applied Bifurcation Theory, Springer, 2004]). From the numerical simulations presented in the paper I see that the systems admit osculations, but I do not see any numerical evidence of a bifurcating limit cycle. I suggest the authors do a revision to clarify these issues.Author Response
Attached pdf.

Round 2
Reviewer 2 Report
From the presented study I still do not see an evidence of
existence of a limit cycle in the paper.
For this reason I recommend to reject the paper.
Author Response
Thanks. See attached our response.

Round 3
Reviewer 2 Report
The paper is improved.In my opinion it can be published.